# The use of brain functional magnetic resonance imaging to determine the mechanism of action of gabapentin in managing chronic pelvic pain in women: a pilot study

Marta Seretny,[1] Sarah Rose Murray,[2] Lucy Whitaker,[2] Jonathan Murnane,[3] Heather Whalley,[4] Cyril Pernet,[3] Andrew W Horne[2]

[1]Edinburgh Cancer Research UK Centre, University of Edinburgh, Edinburgh, UK
[2]MRC Centre for Reproductive Health, University of Edinburgh, Edinburgh, UK
[3]Queens Medical Research Institute and Edinburgh Imaging Facility (QMRI), University of Edinburgh, Edinburgh, UK
[4]Centre for Clinical Brain Sciences, University of Edinburgh, Edinburgh, UK

**Correspondence to**
Dr Andrew W Horne;
andrew.horne@ed.ac.uk

## ABSTRACT

**Objective** To inform feasibility and design of a future randomised controlled trial (RCT) using brain functional MRI (fMRI) to determine the mechanism of action of gabapentin in managing chronic pelvic pain (CPP) in women.

**Design** Mechanistic study embedded in pilot RCT.

**Setting** University Hospital.

**Participants** Twelve women (18–50 years) with CPP and no pelvic pathology (follow-up completed March 2014).

**Intervention** Oral gabapentin (300–2700 mg) or matched placebo.

**Outcome measures** After 12 weeks of treatment, participants underwent fMRI of the brain (Verio Siemens 3T MRI) during which noxious heat and punctate stimuli were delivered to the pelvis and arm. Outcome measures included pain (visual analogue scale), blood oxygen level dependent signal change and a semi-structured acceptability questionnaire at study completion prior to unblinding.

**Results** Full datasets were obtained for 11 participants. Following noxious heat to the abdomen, the gabapentin group (GG) had lower pain scores (Mean: 3.8 [SD 2.2]) than the placebo group (PG) (Mean: 5.8 [SD 0.9]). This was also the case for noxious heat to the arm with the GG having lower pain scores (Mean: 2.6 [SD 2.5]) than the PG (Mean: 6.2 [SD 1.1]). Seven out of 12 participants completed the acceptability questionnaire. 71% (five out of seven) described their participation in the fMRI study as positive; the remaining two rated it as a negative experience.

**Conclusions** Incorporating brain fMRI in a future RCT to determine the mechanism of action of gabapentin in managing CPP in women was feasible and acceptable to most women.

**Trial registration number** ISRCTN70960777.

## Strengths and limitations of this study

► This pilot study includes women with chronic pelvic pain (CPP) randomised to receive placebo or gabapentin for its management.
► All participants in this pilot study underwent a brain functional MRI (fMRI) scan.
► This study estimated sample size for a larger fMRI study assessing gabapentin in CPP and also assessed study acceptability.
► The fMRI sample was too small to draw any statistical conclusions, all results are suggestive of trends only.

## INTRODUCTION

Chronic pelvic pain (CPP) affects >3.3 billion women worldwide and costs an estimated >$225 million/year (£150 million) in socioeconomic costs in the UK alone.[1] Underlying pathology, such as endometriosis or adhesions, can be identified in around 65% of women with CPP. In the remaining 35% of women, pain symptoms have no clear aetiology.[2 3] It is increasingly recognised that women with CPP of unknown aetiology likely represent a subgroup of patients who have undergone central sensitisation (CS), an increased responsiveness of central nociceptive neurons to their normal or subthreshold afferent input.[4] The treatment for women with CPP of unknown aetiology is challenging.[5]

Currently, there are no evidence-based analgesics for CPP,[6] but gabapentin (a GABA analogue, eliciting its analgesic effects via the spinal N-type calcium channel[7]) is increasingly prescribed due to its known effectiveness in other neuropathic pain states.[6] The true efficacy, mechanism of action and side effect profile of gabapentin in women with CPP remain unknown. However, in general, investigation of analgesic drug efficacy in clinical trials is not straightforward. Due to the subjective nature of pain and the

influence of an active physiological placebo response, trials investigating analgesics, even when appropriately powered and carefully executed, often produce small effect sizes that are difficult to interpret clinically.[8] Nevertheless, the increase in the use of adjuvant research tools, such as brain functional MRI (fMRI), has been shown to strengthen and mechanistically explain findings.[9]

We therefore undertook a pilot exploratory study to determine the feasibility of investigating the mechanism of action of gabapentin in managing women with CPP using brain fMRI. The objectives of the study were as follows:

1. To determine whether it is possible to recruit women with CPP, who had agreed to participate in a related pilot clinical trial and received gabapentin or placebo, to partake in an embedded mechanistic study and undergo a brain fMRI scan.
2. To analyse the fMRI brain data and use it for sample size calculation for a future large fMRI brain study investigating the mechanism of action of gabapentin in managing CPP in women.
3. To assess if the fMRI and associated noxious stimuli were acceptable to participants.

## MATERIALS AND METHODS

### Study design

This was a pilot exploratory mechanistic study. The flow of patients in the study is detailed in a Consolidated Standards of Reporting Trials diagram (see figure 1).

### Setting

Women with CPP in NHS Lothian (UK), already recruited as part of a pilot randomised controlled trial (RCT) conducted in NHS Lothian and NHS Grampian (UK) to investigate the efficacy of gabapentin versus placebo in CPP management,[10 11] were approached and invited to participate in this embedded fMRI substudy. The pilot RCT aimed to enrol 60 participants over a 9-month period. The brain fMRI scans were carried out on a Siemens Verio 3T MRI Scanner located at the Edinburgh Imaging Facility, Queens Medical Research Institute (previously known as the Edinburgh Clinical Research Imaging Centre), at the University of Edinburgh (UK). A 12 channel receiver head coil was used. Other study measures were tested in NHS Lothian clinical research facilities or, where appropriate, via the telephone.

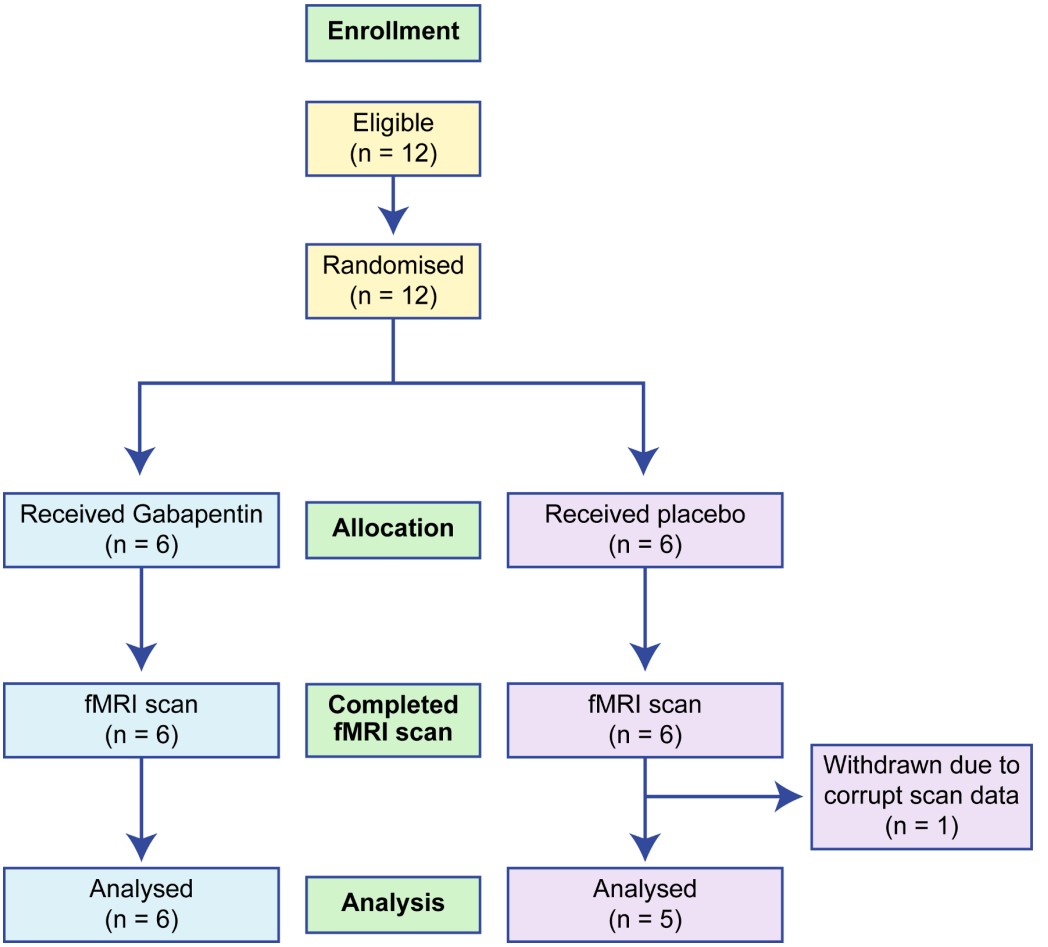

**Figure 1** CONSORT diagram. CONSORT, Consolidated Standards of Reporting Trials; fMRI, functional MRI.

## Inclusion criteria

► Women aged between 18 and 50 years.
► CPP (non-cyclical pain with or without dysmenorrhoea or dyspareunia) assessed clinically to be located within the true pelvis or between and below anterior iliac crests.[12]
► Associated functional disability.
► No obvious pelvic pathology at laparoscopy (<36 months and >2 weeks ago).
► Normal pelvic ultrasound (<36 months).
► Using effective contraception, for example, barrier methods, combined contraceptive pill/oral contraceptives and so on.
► Written informed consent.

## Exclusion criteria

► Known pelvic pathology (eg, endometriosis, cyst, fibroid >3 cm).
► Past history of having taken gabapentin or pregabalin.
► Due to undergo surgery in the next 6 months.
► History of significant renal impairment.
► Allergic to gabapentin or excipients.
► Breast feeding.
► Pregnancy or planning pregnancy in the next 6 months.
► Metal implant/pacemaker/claustrophobia (fMRI subgroup).

## Intervention

Patients were randomised to receive gabapentin at an increasing dose (300–2700 mg) or identical placebo tablet. Gabapentin was started at 300 mg daily and titrated up weekly, in 300 mg increments or until the patient perceived a 50% reduction in pain, or had side effects, to a maximum of 2700 mg. Full details are published in the related clinical trial protocol and results.[10 11]

## Outcome measures

Women underwent an fMRI scan of the brain after a minimum of 12 weeks of treatment, to ensure that all participants had reached their steady state dosing regimen at the time of scanning. The following sequences were collected: structural MPRAGE sequence, resting state blood oxygen level dependent (BOLD) sequence where participants were asked to fixate on a cross, BOLD sequences during which punctate and noxious heat stimuli were applied to the arm and lower abdomen, respectively. Full details of MRI acquisition parameters are shown in table 1. The order of the presented stimuli was randomised (some participants had thermal stimulation applied to the arm then abdomen, while others had it applied in the reverse order). Thermal stimulus was individually thresholded outside the scanner for each participant. For the arm, a MEDOC Pathway thermode was placed on the inner arm, three fingerbreadths proximal to the wrist joint. Temperature was increased from 30°C until the patient reported a pain score of 5 out of 10. The same was repeated for the lower abdomen (three finger breadths above the pubic bone) until the patient reported a pain score of 5 out of 10. These temperatures were used in the scanner in a block experimental design

| Table 1 | MRI acquisition parameters | |
| --- | --- | --- |
| **Weighting** | **T1 weighted image** | **Functional MRI (T2*)** |
| Pulse sequence | Inversion recovery | Fast gradient echo |
| Imaging type | MPRAGE | Echo plannar imaging |
| Flip angle | 9 | 90 |
| Echo time (TE) | 2.98 ms | 30 ms |
| Repetition time (TR) | 2.3 s | 2.5 s |
| Number of volumes | n/a | Thermal runs: 245 Punctate run: 260 Resting state: 200 |
| Number of discarded volumes | n/a | 0 |
| Phase encoding | Anterior/Posterior | Anterior/Posterior |
| Parallel imaging | GRAPPA *2 | GRAPPA *2 |
| Slice order and timing | Interleaved | Interleaved |
| Slice position procedure | ACPC | ACPC |
| Brain coverage | Full brain | Full brain |
| Field of view | 256 | 192 |
| Matrix size | 256 * 256 | 64*64 |
| Slice thickness | 1 mm | 3 |
| Prospective motion correction | None | On |

Details provided follow recommendations of the COBIDAS report (http://biorxiv.org/content/early/2016/07/10/054262).
ACPC, Anterior Commissure Posterior Commisure.

**Table 2** Semi-structured study acceptability questionnaire

| Regarding the fMRI substudy | Positive | | | Negative | |
|---|---|---|---|---|---|
| | 1 | 2 | 3 | 4 | 5 |
| A. … was the choice to opt-in or out a positive or negative one? Did you feel pressured? | | | | | |
| B. … did you feel you were fully informed as to why you were having it? | | | | | |
| C. … was the fMRI a comfortable experience? | | | | | |
| D. … how did you find having the blood sample taken? | | | | | |
| E. … experiences and thoughts on the pain threshold exam | | | | | |

(temperature at neutral for 10 s and then increased to the participant's pre-established noxious heat threshold and held for 15 s and then returned to neutral). The duration of the thermal blocks was 10 minutes on the arm and 10 minutes on the lower abdomen. Pain scores were recorded in real time during the application of the thermal stimulus using a visual analogue scale presented to participants on a screen inside the scanner. Each participant used an automated pointer to rate her pain. Standardised punctate stimuli were applied to the lower abdomen, three fingerbreadths above the pubic bone using a 300 g Touch Test von Frey filament (6.65 mm) in an event-related experimental design. The full duration of the punctate experiment was 10 minutes.

To assess the acceptability of the fMRI substudy, participating women were invited to complete a semi-structured questionnaire administered by telephone at the conclusion of the study protocol prior to unblinding. The questions were designed to assess the participant's feelings and experiences related to the fMRI scan (table 2). The responses were recorded on a structured coding sheet. Respondents were asked to answer each question using a five-point positive (one) to negative (five) scale, with the option of adding free comments.

Women were asked these questions during a phone interview taking place prior to study unblinding. The researcher recorded free comments elaborating each question.

### Statistical analysis

All statistical analyses were carried out prior to participant and clinician unblinding. FMRI statistical analysis was carried out using Statistical Parametric Mapping (SPM12) software.[13] fMRI data first underwent a slice timing correction (interpolation based on a fast Fourier transform, 45th slice as reference), followed by motion correction (6° affine registration minimising the mean square difference, realigned to the first scan of each session then to the mean, fourth degree B-spline interpolation). The T1 weighted image was enhanced using SUSAN[14] from FMRIB Software Library. Images were then coregistered to the mean EPI image (12° affine transformation maximising the normalised mutual information with no interpolation applied) and then simultaneously segmented/bias corrected to derive normalisation parameters to the Montreal Neurological Institute (MNI)

space.[15] Volume-based inter-subject registration was next performed applying the normalisation parameters to all EPI images (fourth degree B-spline interpolation, final voxel size [2 2 2]). These images were finally smoothed with a 6 mm isotropic Gaussian kernel. For the first-level analysis, the stimuli of each session were modelled convolving stimulus onsets with a double gamma haemodynamic response function with derivates[16] and durations of 0 s. Design matrices also included standard six motion parameters and filters for low-frequency (high pass filter 128 s) and high-frequency noise (first-order auto-regressive plus Gaussian whitening). For the thermal stimuli, presentation scales were also modelled for design completeness but not investigated. Regions of interest (ROIs) were defined using NeuroSynth[17] with the keywords chronic pain (http://neurosynth.org/analyses/terms/chronic_pain/). The mean beta values among voxels in 6 mm spheres centred on the prefrontal thalamus ([±10−15 4]), the insulae (Iq2 [±37−20 14]), the periaqueductal gray (PAG - [0 23−17]) and whole brain stem ((0−39 54]). These were obtained for each subject, and group comparison was performed for each ROI (percentile bootstrap on median difference, p=0.05 Bonferroni corrected[18]). Power calculations to calculate the sample size needed for 80% power at a significance level of p=0.05 were conducted using the GPower V.3.1 program.[19] The entire dataset was prepared following the Brain Imaging Data Structure[20] and is available on the University of Edinburgh DataShare repository.[21] Statistical maps are available on NeuroVault[22] (https://neurovault.org/collections/5029/).

### Qualitative analysis

Questionnaire data were assessed thematically in order to identify participant's feelings and experiences related to the fMRI scan and associated noxious heat thresholding. Free comments were reviewed to gain greater understanding on themes important to individual women in the study.

### Ethical approval

All participants gave written informed consent prior to taking part in this study. Consent was also reconfirmed verbally before every fMRI scan.

### Patient and public involvement

The patient co-founder of the Pelvic Pain Support Network helped with the design of this study and also assisted with

**Table 3** Demographics of women who underwent a brain fMRI scan

| Characteristic | Gabapentin (% of 6 or 95% CI) | Placebo (% of 6 or 95% CI) |
|---|---|---|
| Age (years) | 31.7 (20.3 to 43.0) | 30.3 (23.1 to 37.5) |
| Pain score (0–10) | 6.6 (5.8 to 7.5) | 5.5 (3.3 to 7.7) |
| BMI (kg/m$^2$) | 25.05 (21.0 to 29.1) | 26.05 (21.2 to 30.9) |
| HADS score (max score 21) | 9.7 (6.6 to 12.7) | 8 (3.4 to 12.5) |
| Parity | | |
| Nulliparous | 4 (66.7%) | 4 (66.7%) |
| Parous | 2 (33.3%) | 2 (33.3%) |
| Higher education received | | |
| High school | 0 (0%) | 1 (16.7%) |
| College/University | 6 (100%) | 5 (83.3%) |

BMI, body mass index; fMRI, functional MRI; HADS, hospital anxiety and depression score.

gathering feedback regarding the proposed study within the charity. A short description of the proposed pilot study, planned resulting RCT and a series of related questions were distributed among members. Encouragingly, from a survey of 85 women involved in the charity, 69% responders said that they would consider taking part in this pilot.

## RESULTS
### Recruitment
One hundred and thirty-seven women fulfilled the eligibility criteria for the main pilot clinical trial, and 47 women were randomised to treatment.[11] Of the 47 women who were randomised, women taking part in the RCT at the Edinburgh University Hospital were offered participation in the fMRI substudy. Twelve consecutive participants were approached to take part in the fMRI substudy. All of these women consented to undergo a brain fMRI scan, completing the full scanning protocol. Their baseline demographics are shown in table 3. One subject (on placebo) was excluded from analysis following discovery of corrupted scan data at the analysis stage of the study. Progression of pain ratings throughout the study, maximum dose of gabapentin achieved, time to reach maximum dose and reported side effects are summarised in table 4.

### fMRI analysis
fMRI brain scan data for six subjects receiving gabapentin and five receiving placebo were analysed. Average pain ratings during the fMRI scan following noxious stimuli of the abdomen were lower for the gabapentin group (GG) (Mean: 3.8 [SD 2.2]) than the placebo group (PG) (Mean: 5.8 [SD 0.9]). This was also the case for noxious heat applied to the arm where the GG had lower

**Table 4** Pain score (0–10 scale) changes during treatment, maximum treatment dose as well as reported side effects

| ID | Group | Baseline pain | 3 mo pain | 6 mo pain | Max dose (mg) | Time to max dose | Side effects noted |
|---|---|---|---|---|---|---|---|
| 0104 | Active | 7 | 4 | 3 | 1800 | 6 | Dry mouth, itchy skin |
| 0107 | Active | 7 | 6 | 4 | 2100 | 6 | Tired |
| 0113 | Placebo | 6 | 6 | 4 | 600 | 8 | Tired, dizzy, nightmares |
| 0114 | Placebo | 6 | 4 | 6 | 2400 | 8 | Dizzy, nausea |
| 0116 | Placebo | 7 | 4 | 5 | 2400 | 12 | Dizzy, nausea |
| 0119 | Placebo | 5 | 4 | 6 | 300 | * | Dizzy, nausea |
| 0120 | Placebo | 5 | 2 | 3 | 600 | 1 | Tired, dizzy |
| 0122 | Placebo | 2 | 1 | 2 | 2100 | 15 | None |
| 0123 | Active | 8 | 7 | 6 | 2400 | 12 | Restlessness |
| 0124 | Placebo | 8 | 7 | 5 | 1800 | 12 | Low mood, nausea |
| 0125 | Active | 6 | 3 | 2 | 1800 | 8 | Dizzy, nausea |
| 0129 | Active | 6 | 4 | 3 | 1800 | 7 | None |

ID represents anonymised study ID number and Active represents gabapentin recipients. Time to max dose is given in weeks.
*Data missing.

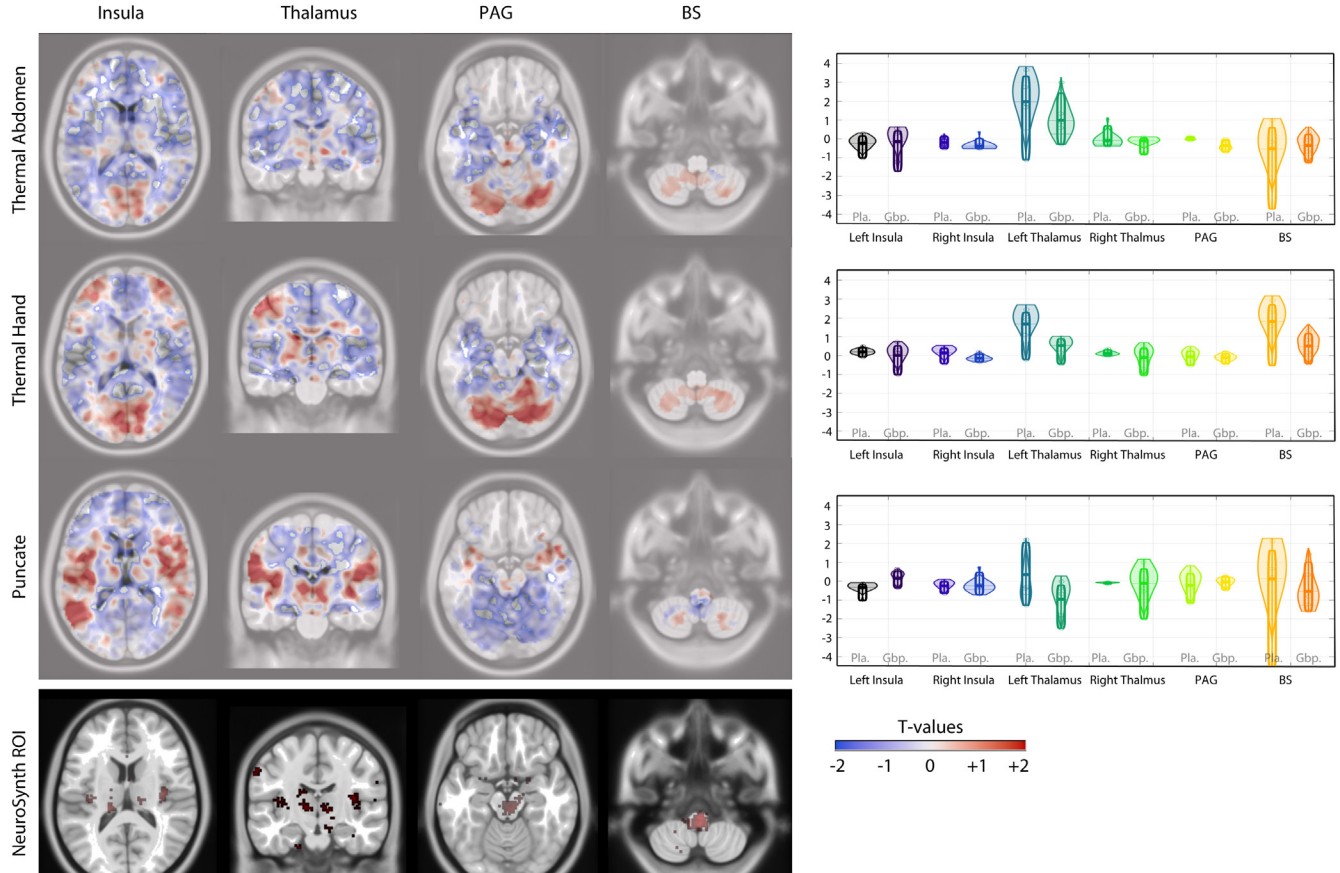

**Figure 2** Activation maps (unthresholded) for each stimulus presented, alongside ROI group comparisons (Available at https://drive.google.com/open?id=1Iroa8uxXLQ-w6-kBPbVtqPagLjTjtuPu). ROI, region of interest.

average pain scores (Mean: 2.6 [SD 2.5]) than the PG (Mean: 6.2 [SD 1.1]). Figure 2 shows unthresholded activation maps for each of the presented stimuli as well as ROIs and the statistical comparison between these. To summarise, the ROI analysis showed that BOLD signal changes in the left PI following punctate stimuli were significantly different between those receiving placebo and those receiving gabapentin (figure 2); this was also the case for activation in the PAG following heat application to the lower abdomen. All other ROI comparisons showed no statistically significant differences (tables 5–7).

As expected, due to the exploratory sample size, group-level whole brain analysis did not yield any statistically significant between group differences.

A sample size calculation from these results suggest that for a mechanistic substudy embedded in a larger double-blind RCT investigating the efficacy of gabapentin versus placebo in CPP, a minimum of 7 women per group (a total of 14 brain fMRI scans) are required for 80% power and p=0.05 significance level, sampling ratio 1:1, with two independent samples and two-sided testing. This calculation was based on differences in BOLD signal activation

**Table 5** ROI median beta values with 95% highest density intervals following thermal stimulation to the abdomen

| Region | Gabapentin | Placebo | Difference | P value |
|---|---|---|---|---|
| Left insula | −0.14 (−1.71 to 0.44) | −0.23 (−1.01 to 0.14) | (−0.98 to 0.82) | 0.8820 |
| Right insula | −0.38 (−0.52 to 0.01) | −0.19 (−0.51 to 0.13) | (−0.54 to 0.32) | 0.5160 |
| Left thalamus | 0.97 (−0.28 to 2.44) | 1.97 (−1.12 to 3.31) | (−2.53 to 1.4) | 0.3620 |
| Right thalamus | −0.11 (−0.83 to 0.03) | −0.08 (−0.37 to 0.66) | (−0.84 to 0.23) | 0.7360 |
| PAG | −0.35 (−0.7 to −0.004) | 0.0004 (−0.07 to 0.09) | (−0.61 to −0.05) | 0.004 |
| Brain stem | −0.3579 (−1.25 to 0.21) | −0.52 (−3.71 to 0.61) | (−1 2.5) | 0.7900 |

Please note difference=percentile bootstrap median difference and associated p value of this.
ROI, region of interest.

**Table 6** ROI median beta values with 95% highest density intervals following thermal stimulation to the hand

| Region | Gabapentin | Placebo | Difference | P value |
|---|---|---|---|---|
| Left insula | 0.01 (−1 to 0.52) | 0.19 (−0.07 to 0.4) | (−0.95 to 0.25) | 0.454 |
| Right insula | −0.13 (−0.33 to 0.1) | 0.14 (−0.42 to 0.39) | (−0.55 to 0.22) | 0.254 |
| Left thalamus | 0.53 (−0.44 to 0.88) | 1.68 (−0.2 to 2.34) | (−2.01 to 0.1) | 0.07 |
| Right thalamus | −0.09 (−1.03 to 0.4) | 0.13 (−0.01 to 0.3) | (−0.91 to 0.21) | 0.388 |
| PAG | −0.11 (−0.42 to 0.1) | −0.05 (−0.5 to 0.29) | (−0.42 to 0.36) | 0.84 |
| Brain stem | 0.5 (−0.42 to 1.16) | 1.8 (−0.52 to 2.72) | (−2.23 to 0.28) | 0.096 |

Please note difference=percentile bootstrap median difference and associated p value of this.
ROI, region of interest.

in the left insula (punctate) and PAG (abdomen thermal stimuli) or if averaging across all ROI and all conditions.

### fMRI substudy acceptability

All 12 women were approached for the telephone interview. Seven were contactable and gave consent to the telephone-administered semi-structured acceptability questionnaire. Five women rated the fMRI as a positive experience, scoring it as 1 or 2 on a 1–5 point, positive to negative scale. Two participants scored their experience as '4' on the scale. One of these commented that the trial substudy process was not explained clearly enough. This participant was also disappointed by lack of feedback after the scan. The other participant rating the experience negatively felt that an in-person explanation rather than the information leaflet would have been clearer. All participants felt the option of opting in or out of the mechanistic fMRI substudy was positive. The majority of women found the experience of the actual scan positive or neutral, with similar ratings given to the quality of information given before the scan, as well as their experience of the noxious pain thresholding. One participant noted that prior to the study, she had anxiety related to MRI scans and that study participation actually helped her overcome her fear of MRI. Only one participant felt that the information given before the scan, the actual scan experience as well as the noxious thresholding was a negative experience for her (a score of 4 on the 5-point scale).

## DISCUSSION
### Recruitment

Our study met our objectives in relation to establishing feasibility of recruitment. All of the participants approached for brain fMRI consented to this part of the study and completed the full hour-long protocol in the scanner. Failure to recruit to trials is known to be a major limitation in clinical research, prompting new recommendations aimed at improving approaches to recruitment.[23] Assessment of study feasibility is a part of this improvement process. Pilot studies are considered a subtype of formal feasibility assessments.[24] Further, pilot studies are deemed valuable in ascertaining unknown factors such as recruitment to novel outcome measures, such as in our case, brain fMRI in CPP patients.[25] We feel our successful recruitment to this study supports future fMRI research investigating the mechanism of action of gabapentin for the management of women with CPP.

### fMRI analysis

The key outcome of our fMRI data analysis was calculation of sample size for a planned future brain fMRI study to investigate the mechanism of action of gabapentin in the management of women with CPP. Sample size calculations for fMRI studies have become a standard part of good MRI research methodology.[26] We also explored BOLD signal changes in response to punctate and noxious heat stimuli, in regions of the brain known to be involved

**Table 7** ROI median beta values with 95% highest density intervals following punctate stimulation to the lower abdomen

| Region | Gabapentin | Placebo | Difference | P value |
|---|---|---|---|---|
| Left insula | 0.14 (−0.35 to 0.54) | −0.33 (−1 to −0.1) | (0.1 to 1.07) | 0.042 |
| Right insula | −0.24 (−0.71 to 0.26) | 0.25 (−0.65 to −0.018) | (−0.3 to 0.6) | 0.892 |
| Left thalamus | −0.95 (−2.5 to −0.14) | −0.34 (−1.28 to 1.99) | (−3.45 to 0.2) | 0.114 |
| Right thalamus | −0.11 (−1.99 to 0.72) | −0.09 (−0.14 to −0.06) | (−1.1 to 0.7) | 0.948 |
| PAG | −0.05 (−0.45 to 0.16) | −0.21 (−1.16 to 0.39) | (−0.4 to 0.99) | 0.654 |
| Brain stem | −0.54 (−1.59 to 1.01) | 0.12 (−4.63 to 1.62) | (−2.1 to 2.49) | 0.65 |

Please note difference=percentile bootstrap median difference and associated p value of this.
ROI, region of interest.

in pain processing. Our findings suggest between group differences in response to punctate and abdominal heat stimuli in the insula and PAG. These trends fit with what is known about the fundamental role of the insula and the PAG in CS and pain processing.[27 28] Specifically, in CPP, both the PAG and insula are known to have altered structure and connectivity.[29–31] Moreover, CS is postulated to be an important mechanism in the maintenance of CPP.[32 33] Neuromodulators such as gabapentin have been shown in neuroimaging studies to modify CS pain states.[34] Further work in a larger mechanistic fMRI study is needed to confirm and elucidate these findings.

## Study acceptability

Patient engagement in clinical trials is known to improve study recruitment and retention.[35] Specific to pain research and analgesic RCTs, many issues limit patient participation in pain trials using fMRI, not least the actual experience of chronic pain.[36] Decreased participation of subsets of pain patients in pain medication trials can introduce potential biases into study findings.[37] Understanding patient experiences, which in our case helped highlight the need for clearer communication around study logistics, will help use fMRI in future mechanistic probing of gabapentin efficacy in CPP. This is particularly valuable, as the utility of fMRI in chronic pain management research is deemed to have growing importance.[38]

## Study strengths and limitations

The major limitation of our fMRI data analysis lies in our small sample size. Approaches to brain fMRI analysis are varied and may be prone to false-positive or non-reproducible results.[39] This is particularly true when small datasets are used.[40] It is therefore possible that a different approach to the analysis of our data may have yielded different results. We therefore interpret these findings as interesting trends, in need of further validation, and we have made our fMRI dataset available to interested researchers who may wish to re-analyse our data or combine it with another relevant dataset. Another limitation of our study is lack of data regarding the duration of CPP in study participants; we have corrected this in the ongoing GaPP2 study. The strengths of this study relate to all the study aims being fulfilled and clearly reported. This study is one of the first to use fMRI in the assessment of women with CPP receiving gabapentin or placebo.

## CONCLUSION

We have demonstrated that it is possible to recruit women with CPP participating in a pilot RCT of gabapentin versus placebo, to take part in a mechanistic substudy involving a brain fMRI scan. Most participants in our pilot found the MRI scan, including administration of noxious stimuli, acceptable. BOLD signal changes identified in our analysis are likely useful guides for further studies aimed at understanding the mechanism of action of gabapentin in CPP. Sample size calculations resulting from this work

have been used to inform the design of a mechanistic fMRI substudy in a larger RCT investigating gabapentin versus placebo for the management of CPP in women.

**Acknowledgements** We thank Katy Vincent (DPhil, BSc, MBBS, MRCOG) who was instrumental in guiding the set-up and data analysis for this study. Thanks to Scott Semple (PhD) and Neil Roberts (PhD), MRI physicists, for their assistance in setting up additional data acquisition equipment and monitoring in the MRI scanner. We would also like to thank Ms Jennifer Brawn (MSc, DPhil) for organising scan sequence transfer and set-up between Oxford and Edinburgh. We thank all the radiographers at the Clinical Research Imaging Centre (CRIC) Edinburgh, for their contribution to data collection. We also thank Mrs Ann Doust and Mrs Helen Dewart for organising patient recruitment and follow-up. Additional thanks are owed to Ann Doust for managing all study data and answering endless queries related to writing this manuscript. We would like to thank Judy Birch JB, the patient liaison and representative of the Pelvic Pain Support Network for her advice and input into this study. Finally, and perhaps most importantly, we would like to thank all the women who took part in this study.

**Contributors** MS, AWH and CP were involved in the design and set-up of the study. MS, LW, SRM, JM and CP were involved in scan data collection. CP analysed the fMRI data. CP and HW performed the sample size calculation. All authors contributed to the writing of this manuscript.

**Funding** This work was supported by a grant from the Chief Scientist's Office Scotland (CZH/4/688) and an MRC Centre Grant Mr/N022556/1. MS was funded by the Wellcome Trust via the Scottish Translational Medicine and Therapeutics Initiative (STMTI). JM held a PhD studentship funded by the Mentholatum company.

**Competing interests** The funders did not have any influence on the study design, data collection, analysis or interpretation of results.

**Patient consent for publication** Not required.

**Ethics approval** The Scotland A Research Ethics committee gave approval for this study (REC 12/SS/0005) on 13 November 2012.

**Provenance and peer review** Not commissioned; externally peer reviewed.

**Data sharing statement** All data from this study are currently held securely at the University of Edinburgh. Anonymised fMRI data are available to interested researchers and can be accessed at (https://datashare.is.ed.ac.uk/handle/10283/3153).

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
