## [Reviewer comments · BMJ Open]

ARTICLE DETAILS

TITLE (PROVISIONAL)	The use of brain functional magnetic resonance imaging to determine the mechanism of action of gabapentin in managing chronic pelvic pain in women: a pilot study
AUTHORS	Seretny, Marta; Murray, Sarah; Whitaker, Lucy; Murnane, Jonathan; Whalley, Heather; Pernet, Cyril; Horne, Andrew

VERSION 1 - REVIEW

REVIEWER	Catherine Jutzeler University of British Columbia, Canada
REVIEW RETURNED	22-Oct-2018

GENERAL COMMENTS	Seretny et al present an interesting study investigating use of brain functional magnetic resonance imaging to determine the mechanism of action of gabapentin in women suffering from chronic pelvic pain. This is an important study as the true efficacy of gabapentin to treat chronic pelvic pain is unknown. The authors enrolled 12 female participants, 6 of which received gabapentin and the remaining 6 placebo. The authors conclude that the incorporation of brain fMRI in future RCTs is feasible in patients with CPP. The study is well-written. I have some comments that should be addressed. • The authors mention that Gabapentin is a GABA analogue. While this is true, I suggest to mention that Gabapentin does, however, not bind to GABA receptors or affect the production or uptake of GABA. To clarify the mechanism of gabapentin is independent of GABA.• The authors state as one inclusion criteria 'Chronic pelvic pain (non-cyclical pain with or without dysmenorrhoea or dyspareunia) located within the true pelvis or between and below anterior iliac crests.' How was the presence of chronic pelvic pain assessed? Please elaborate.• When describing the intervention, the authors refer to reader two published papers that provide all the details. However, it would make it easier for the reader, if the authors would briefly explain the titration procedures. This is just a suggestion and I leave it up to the authors if they wanna change this.• What is the rationale to scan the patients six weeks after treatment?
--

• The authors state that they used SPM8. However, there is a newer version of SPM (Version 12, it is freely available). Please explain why the authors chose to use SPM8.

• The fMRI analysis was false discovery rate, which is the less stringent correction for multiple comparison when using fMRI. I strongly suggest to the authors to use Family Wise Error (FWE) correction, particularly in such as small sample.

FWE, the family-wise error rate, is the probability of making at least one error ($V > 0$) $P(V > 0)$.

FDR, the false discovery rate, is the expected value ratio of errors we have made: $E(V/\max(R, 1))E(V/\max(R, 1))$.

• The authors did not state if any covariates were included in the fMRI analysis, such as age, CPP pain intensity, dose of gabapentin? These are important co-variables that are known to affect the fMRI results. Please elaborate and add the details.

• Please provide details on how the sample size was calculated. In the methods section, the authors simply state that the sample size calculation was performed. Since this is an integral part of the study, more details should be provided.

• Please refrain from using the term 'trended towards significance ($p=0.051$), unless you term $p=0.049$ 'trend towards insignificance'.

• In Table 2, can the authors add more information about the CPP: Duration of pain, intensity of pain, etc? These are important information.

• The authors should provide data on the treatment: Dose, time until max dose was reached, side effects. This information should also be stated for the placebo group. A table with this information would be helpful.

• In the discussion, the authors state 'We found a trend suggesting decreased pain processing in the posterior insula in women who were receiving gabapentin compared those who were receiving placebo.' Considering the small number of patients and the fact that the result was not significant, the authors should be careful with their conclusions regarding the mechanism of gabapentin.

• Did the authors look at correlation between the fMRI activation and the pain relief? Patients who experience more pain relief might show higher activation.

• The authors should present some brain images with the BOLD activation interlaced. While figure 1 is informative, it would be important to show the 'raw data' to get an impression of the actual BOLD activation.

• Line 51/ page 16: There is a spelling error 'tend' instead of 'trend'.

• The patients were asked about the tolerability of fMRI scans. Did the authors also consider to ask the patients about the gabapentin treatment? It is known that gabapentin can have major side effects and that some patients don't tolerate these or prefer to switch to another pain medication.

REVIEWER	Dr. ANANDH KILPATTU RAMANIHARAN THE UNIVERSITY OF ALABAMA AT BIRMINGHAM
REVIEW RETURNED	13-Jan-2019

GENERAL COMMENTS	The authors have presented a pilot study with good inferences. In the methodology, the authors have to mention about the protocol details about the MR imaging. What are the values for TE/TR in the sequences and any other associated parameter values. This would be helpful for the readers to repeat the experiment in future.
---

REVIEWER	Sawsan As-Sanie University of Michigan USA
REVIEW RETURNED	02-Mar-2019

GENERAL COMMENTS	The authors present a pilot exploratory study which successfully demonstrates the feasibility of using fMRI to investigate the mechanism of action of gabapentin in the treatment of chronic pelvic pain. This is a very well-written manuscript which provides critical information in the feasibility, acceptability and design of neuroimaging studies in women with chronic pelvic pain. I have very few comments because this study is well-designed and the manuscript is well-written. The following comments are minor, but addressing them would improve the interpretation of the findings and the quality of the manuscript.  1. How were 12 participants randomly selected for fMRI? I assume that recruitment accrued over time, and at the time of enrollment, there was no guarantee that the study would recruit adequate numbers. Were all participants offered fMRI until reaching target enrollment numbers? Was a random number generator used to select 12 out of the 47 women who were enrolled in this pilot study? 2. Although the authors provide reference to the pilot gabapentin study, it would be helpful to briefly summarize the key features of the protocol. For example, what was the target enrollment for the pilot RCT? What was the actual enrollment? I have heard that a larger study is underway- how does this pilot study contribute to the full protocol? 3. The authors conducted a telephone interview to assess patient attitudes and acceptability of completing this protocol and undergoing fMRI. Since there are only 12 participants who completed fMRI, why not invite all 12 participants to complete telephone interview regarding acceptability? How was the sample size of 7 interviews selected? How were these 7 women "randomly" selected? 4. Please provide the QST data (pre-post gabapentin) in a table format. 5. Typo, page 16, line 52, tends=trends 6. Please check the format of the references. Many of the articles do not have full page numbers. 7. The authors indicate that there the insula is known to have altered structure and connectivity in CPP- but reference 19 is a review article. Why not provide reference to the original papers that actually demonstrate this finding in CPP. Similarly, central sensitization has been demonstrated to be present in women with
--

	CPP, with demonstrated differences in QST and central nervous system structure and function. Again the authors provide an important review article (ref 20) as a reference, but do not reference the original work.
--	---

VERSION 1 – AUTHOR RESPONSE

Reviewer #1

We are encouraged that Dr Jutzeler thought that our study was interesting and well-written, and that she agrees with our assessment of a need for more information on this important clinical problem.

COMMENT: The authors mention that Gabapentin is a GABA analogue. While this is true, I suggest to mention that Gabapentin does, however, not bind to GABA receptors or affect the production or uptake of GABA. To clarify the mechanism of gabapentin is independent of GABA.

RESPONSE: We have amended the introduction accordingly, on page 5.

COMMENT: The authors state as one inclusion criteria 'Chronic pelvic pain (non-cyclical pain with or without dysmenorrhoea or dyspareunia) located within the true pelvis or between and below anterior iliac crests.' How was the presence of chronic pelvic pain assessed? Please elaborate.

RESPONSE: This was a clinical assessment and we have amended the methods accordingly, on page 7.

COMMENT: When describing the intervention, the authors refer to reader two published papers that provide all the details. However, it would make it easier for the reader, if the authors would briefly explain the titration procedures. This is just a suggestion and I leave it up to the authors if they want to change this.

RESPONSE: We have amended the methods accordingly, on page 8.

COMMENT: What is the rationale to scan the patients six weeks after treatment?

RESPONSE: This is now explained on page 9 and corrected as the scans were performed after 12 weeks of treatment.

COMMENT: The authors state that they used SPM8. However, there is a newer version of SPM (Version 12, it is freely available). Please explain why the authors chose to use SPM8.

RESPONSE: This was written in error, SPM 12 not 8 was used, and this has been corrected on page 11.

COMMENT: The fMRI analysis was false discovery rate, which is the less stringent correction for multiple comparison when using fMRI. I strongly suggest to the authors to use Family Wise Error (FWE) correction, particularly in such as small sample. FWE, the family-wise error rate, is the probability of making at least one error $P(V>0)$. FDR, the false discovery rate, is the expected value ratio of errors we have made: $E(V/\max(R,1))$.

RESPONSE: Collectively all of these comments and questions regarding our fMRI data analyses rekindled a previously held debate amongst our group related to the best approach to analyse this pilot data, related to approaching data with small sample sizes and defining ROIs. In conclusion, it led us to re-run the whole fMRI analysis. Previously, we had defined our a priori chosen ROIs using data from our Oxford University colleague Dr Katy Vincent's fMRI work on women with pelvic pain (Vincent 2011). In this re-analysis, we defined these regions using metanalyses methods (Yarkoni 2011). There is argument for both approaches, but we felt the metanalyses defined ROIs optimised robustness by being derived from many chronic pain fMRI studies. The limitation of this approach is potential loss of condition related specificity (i.e. the chronic pain term identifies studies investigating all chronic pain states and not just chronic pelvic pain). We have now also numerically shown all formal ROI group comparisons to avoid any misleading suggestions about statistical significance, which may have arisen out of our previous bar graph. We also give more explicit detail about our sample size calculation methods related to the documented ROI data. These changes are highlighted throughout the text but most notably on pages 11,12,16,17 and 18.

COMMENT: The authors did not state if any covariates were included in the fMRI analysis, such as age, CPP pain intensity, dose of gabapentin? These are important co-variables that are known to affect the fMRI results. Please elaborate and add the details.

RESPONSE: We did not include any covariates in this analysis due to the small sample and aim of calculating samples size related to regions of interest only.

COMMENT: Please provide details on how the sample size was calculated. In the methods section, the authors simply state that the sample size calculation was performed. Since this is an integral part of the study, more details should be provided.

RESPONSE: We have provided specifics in the methods on page 13.

COMMENT: Please refrain from using the term 'trended towards significance ($p=0.051$), unless you term $p=0.049$ 'trend towards insignificance'.

RESPONSE: We have endeavoured to remove any potentially misleading language from our manuscript. The results related to this comment have been removed and replaced with the re-run analyses and numerical summaries for all comparisons made (tables 6,7,8 and online figure 2).

COMMENT: In Table 2, can the authors add more information about the CPP: Duration of pain, intensity of pain, etc? These are important information.

RESPONSE: Unfortunately, we did not collect duration of pain data during this pilot study. We noted this omission and this data is being explicitly recorded in the ongoing definitive RCT resulting from this pilot the GaPP2 study (Vincent 2018). We have added this as a limitation on page 21.

COMMENT: The authors should provide data on the treatment: Dose, time until max dose was reached, side effects. This information should also be stated for the placebo group. A table with this information would be helpful.

RESPONSE: We have added table 4 with max dose, changing pain scores and side effects information on page 15.

COMMENT: In the discussion, the authors state 'We found a trend suggesting decreased pain processing in the posterior insula in women who were receiving gabapentin compared those who were receiving placebo.' Considering the small number of patients and the fact that the result was not significant, the authors should be careful with their conclusions regarding the mechanism of gabapentin.

RESPONSE: We have removed this sentence from the manuscript.

COMMENT: Did the authors look at correlation between the fMRI activation and the pain relief? Patients who experience more pain relief might show higher activation.

RESPONSE: We have not looked at any correlations, please see our response above on keeping the fMRI analysis exploratory and strictly related to sample size calculation only.

COMMENT: The authors should present some brain images with the BOLD activation interlaced. While figure 1 is informative, it would be important to show the 'raw data' to get an impression of the actual BOLD activation.

RESPONSE: We have added this as discussed on page 16 and the figure (2) is available online at <https://drive.google.com/file/d/1Iroa8uxXLQ-w6-kBPbVtqPagLjTjtuPu/view>

COMMENT: Line 51/ page 16: There is a spelling error 'tend' instead of 'trend'.

RESPONSE: Corrected.

COMMENT: The patients were asked about the tolerability of fMRI scans. Did the authors also consider to ask the patients about the gabapentin treatment? It is known that gabapentin can have major side effects and that some patients don't tolerate these or prefer to switch to another pain medication.

RESPONSE: We have added this information to Table 4, page 15.

Reviewer #2

COMMENT: The authors have presented a pilot study with good inferences. In the methodology, the authors have to mention about the protocol details about the MR imaging. What are the values for TE/TR in the sequences and any other associated parameter values. This would be helpful for the readers to repeat the experiment in future.

RESPONSE: We are also encouraged by Dr Kilpattu Ramaniharan's positive feedback and are grateful to her for drawing our attention to our oversight in detailing MR imaging parameters in our manuscript. We have included relevant parameters in table 1 on page 10 of our revised manuscript.

Reviewer # 3

We are grateful that Dr As-Sanie thinks that our manuscript is very well-written and provides critical information in the feasibility, acceptability and design of neuroimaging studies in women with chronic pelvic pain.

COMMENT: How were 12 participants randomly selected for fMRI? I assume that recruitment accrued over time, and at the time of enrolment, there was no guarantee that the study would recruit adequate numbers. Were all participants offered fMRI until reaching target enrolment numbers? Was a random number generator used to select 12 out of the 47 women who were enrolled in this pilot study?

RESPONSE: The selection of fMRI participants is explained on page 14. The word random has been removed as it was consecutive women who were approached. Ultimately, we only had sufficient funding to perform 12 fMRI scans.

COMMENT: Although the authors provide reference to the pilot gabapentin study, it would be helpful to briefly summarize the key features of the protocol. For example, what was the target enrollment for the pilot RCT? What was the actual enrollment? I have heard that a larger study is underway- how does this pilot study contribute to the full protocol?

RESPIONSE: We have included more detail regarding the enrolment to the whole pilot RCT on page 7. The fMRI pilot study described here has contributed to the definitive trial's (GaPP2) fMRI mechanistic sub-study as discussed on page 23. We are reluctant to repeat any other previously published details of the pilot RCT as different authors are involved and the related publication contains the full details.

COMMENT: The authors conducted a telephone interview to assess patient attitudes and acceptability of completing this protocol and undergoing fMRI. Since there are only 12 participants who completed fMRI, why not invite all 12 participants to complete telephone interview regarding acceptability? How was the sample size of 7 interviews selected? How were these 7 women "randomly" selected?

RESPONSE: On the basis of this helpful comment, we went back to our data collection hard copies and saw that all 12 were actually approached but only 7 were contactable. This is clarified on page 18.

COMMENT: Please provide the QST data (pre-post gabapentin) in a table format.

RESPONSE: The scanning occurred at a single time point after 12 weeks of treatment. Therefore, scan related QST (the only QST conducted in this pilot) does not have a pre and post treatment time point.

COMMENT: Typo, page 16, line 52, tends=trends

RESPONSE: Corrected.

RESPONSE: Please check the format of the references. Many of the articles do not have full page numbers.

RESPONSE: All references have been reviewed and corrections made where applicable.

RESPONSE: The authors indicate that there the insula is known to have altered structure and connectivity in CPP- but reference 19 is a review article. Why not provide reference to the original papers that actually demonstrate this finding in CPP. Similarly, central sensitization has been demonstrated to be present in women with CPP, with demonstrated differences in QST and central nervous system structure and function. Again the authors provide an important review article (ref 20) as a reference, but do not reference the original work.

RESPONSER: This has been amended on page 20

VERSION 2 – REVIEW

REVIEWER	Catherine Jutzeler University of British Columbia
REVIEW RETURNED	08-May-2019

GENERAL COMMENTS	The reviewer completed the checklist but made no further comments.
--

REVIEWER	Dr. Anandh Kilpattu Ramaniharan The University of Alabama at Birmingham
REVIEW RETURNED	10-May-2019

GENERAL COMMENTS	The authors may want to increase the number of participants in future.
--

REVIEWER	Sawsan As-Sanie, MD MPH University of Michigan United States of America
REVIEW RETURNED	11-May-2019

GENERAL COMMENTS

The authors have adequately addressed all prior reviewer comments. The manuscript is clearly written and provides a significant contribution to the literature.